# Genetic and Biochemical Characterizations of aLhr1 Helicase in the Thermophilic Crenarchaeon *Sulfolobus acidocaldarius*

**Shoji Suzuki [1], Norio Kurosawa [2], Takeshi Yamagami [1], Shunsuke Matsumoto [1], Tomoyuki Numata [1], Sonoko Ishino [1,\* and Yoshizumi Ishino [1,\***

[1] Department of Bioscience and Biotechnology, Graduate School of Bioresource and Bioenvironmental Sciences, Kyushu University, Fukuoka 819-0395, Japan; s.suzukie@gmail.com (S.S.); yamagami@agr.kyushu-u.ac.jp (T.Y.); smatsumoto@agr.kyushu-u.ac.jp (S.M.); tomoyuki.numata94@agr.kyushu-u.ac.jp (T.N.)

[2] Department of Science and Engineering for Sustainable Development, Faculty of Science and Engineering, Soka University, Hachioji 192-8577, Japan; kurosawa@soka.ac.jp

[*] Correspondence: ishino@agr.kyushu-u.ac.jp (S.I.); sonoko@agr.kyushu-u.ac.jp (Y.I.); Tel.: +81-92-802-4715 (S.I.); +81-92-802-4714 (Y.I.); Fax: +81-92-802-4696 (S.I. & Y.I.)

**Abstract:** Homologous recombination (HR) refers to the process of information exchange between homologous DNA duplexes and is composed of four main steps: end resection, strand invasion and formation of a Holliday junction (HJ), branch migration, and resolution of the HJ. Within each step of HR in Archaea, the helicase-promoting branch migration is not fully understood. Previous biochemical studies identified three candidates for archaeal helicase promoting branch migration in vitro: Hjm/Hel308, PINA, and archaeal long helicase related (aLhr) 2. However, there is no direct evidence of their involvement in HR in vivo. Here, we identified a novel helicase encoded by Saci_0814, isolated from the thermophilic crenarchaeon *Sulfolobus acidocaldarius*; the helicase dissociated a synthetic HJ. Notably, HR frequency in the Saci_0814-deleted strain was lower than that of the parent strain (5-fold decrease), indicating that Saci_0814 may be involved in HR in vivo. Saci_0814 is classified as an aLhr1 under superfamily 2 helicases; its homologs are conserved among Archaea. Purified protein produced in *Escherichia coli* showed branch migration activity in vitro. Based on both genetic and biochemical evidence, we suggest that aLhr1 is involved in HR and may function as a branch migration helicase in *S. acidocaldarius*.

**Keywords:** archaea; homologous recombination; branch migration helicase; hyperthermophile; Holliday junction

## 1. Introduction

Genomic DNA, which encodes genetic information, is always damaged by endogenous and exogenous factors, and the frequency of damage is accelerated by two to three orders of magnitude at high temperatures [1]. Hyperthermophiles are microorganisms that flourish in hot environments (above 80 °C) [2]. The topic of how hyperthermophiles consistently maintain their genome integrity under extreme environments has been discussed in recent years, and the idea that hyperthermophiles efficiently repair DNA damage occurring at elevated levels has been proposed. Thus, studies to elucidate DNA repair mechanisms in hyperthermophiles are necessary for understanding the mechanisms underlying the maintenance of genetic information in living cells. In particular, most hyperthermophiles belong to the Archaea domain [2]; thus, the DNA repair mechanisms in archaeal cells have been studied (reviewed in [3–10]). However, these mechanisms remain to be completely elucidated, and further analyses are required.

In several DNA repair pathways, homologous recombination (HR) is important for both stalled-replication fork and double-strand break repairs in hyperthermophilic Archaea (HA). It serves as the major DNA repair pathway for the removal of a wide variety of DNA



lesions and is unusually efficient and reliable in comparison with that in other organisms [7]. The HR process is generally composed of four steps, and some of the proteins have been predicted to work for each step in HA [9]. The first step is end resection to produce a 3′-single-stranded DNA (ssDNA) by the Rad50-Mre11-HerA-NurA complex. In the second step, the 3′-ssDNA is used for strand invasion and formation of a Holliday junction (HJ), which is catalyzed by recombinase RadA with assistance from accessory proteins. In the third step, branch migration of the HJs is possibly catalyzed by candidates for branch migration helicase. Finally, HJ is resolved by the HJ-specific endonuclease Hjc and/or its paralog Hje in some Archaea.

To date, three proteins, namely, Holliday junction migration (Hjm)/Hel308, PilT N-terminal-(PIN)-domain-containing ATPase (PINA), and archaeal long helicase related (aLhr) 2, have been characterized as candidates for branch migration helicase because these helicases show branch migration activity in vitro. Branch migration activity against a synthetic HJ is defined as the ability to dissociate a synthetic HJ to half junctions (splayed DNA). As the first report of a candidate for branch migration helicase, Hjm from the hyperthermophilic euryarchaeon *Pyrococcus furiosus* has been characterized in vitro, given that it shows branch migration activity against a synthetic HJ and plasmid-based recombinase-mediated recombination intermediate [11]. The Hjm homolog (Hel308) from the euryarchaeon *Methanothermobacter thermautotrophicus* also has a structure-specific helicase activity [12]. Genetic complementation experiments using the *E. coli dnaE* mutant strain demonstrated that the role of Hjm in *P. furiosus* and Hel308 of *M. thermautotrophicus* was identical to that of the RecQ helicase, suggesting that Hjm/Hel308 may be involved in the repair of stalled replication forks [12,13]. Hjm/Hel308 from the crenarchaeon *Sulfolobus tokodaii* (reclassified as *Sulfurisphaera tokodaii* [14]) [15] also have unwinding activity against a synthetic HJ (without product assignment) and a chicken-foot structure produced by replication fork regression, respectively. Guy and Bolt demonstrated that the UV sensitivity of the Δ*ruvABC* or Δ*ruvABC*/Δ*recG* mutant *E. coli* cells increased when the gene for *M. thermautotrophicus* Hel308 was expressed in these strains, suggesting that Hel308 interfered with another repair process independently [12].

The second candidate is PINA, which was originally identified as a novel ATPase associated with Hjc, the HJ resolvase, from the hyperthermophilic crenarchaeon *Saccharolobus* (formerly *Sulfolobus*) *islandicus* [16]. PINA forms a hexameric ring, promotes branch migration of a synthetic HJ in vitro, as observed for the bacterial branch migration helicase RuvB [17]. This report also proposed a mechanism for branch migration by PINA, and further analysis proposed mechanisms of stalled replication fork repair based on the functions of Hjm, PINA, and Hjc [18]. Based on these results, *S. islandicus* PINA appears to be a helicase involved in the HJ branch migration process. However, its physiological role in vivo needs to be further characterized.

The third candidate was the protein named long helicase related (Lhr), which was originally identified and characterized in bacteria [19,20]. The first characterized aLhr is SSO0112 from *Saccharolobus solfataricus* [21], which was later identified again as Saci_1500 from *Sulfolobus acidocaldarius* [22]. This helicase can unwind a synthetic HJ, thereby producing ssDNA, but it cannot produce half junctions. The mutant strain with deletion of the corresponding gene is more sensitive to UV light but not to hydroxyurea and methyl methanesulfonate (MMS) [22], suggesting that this group of helicases function in some specific pathway including the UV damage repair in *Sulfolobales*. The recombination frequencies were not different between the wild type and mutant strains, suggesting that the function of this helicase is not important for the HR process in vivo. More recent studies showed that Mt_1802, from *M. thermautotrophicus*, catalyzes the branch migration of a synthetic HJ and compensates for the loss of RecQ function in *E. coli* cells. These properties are similar to those of Hjm/Hel308 [23]. A comprehensive phylogenetic analysis of the superfamily 2 (SF2) helicases in living organisms proposed that aLhrs can be divided into two groups: Lhr1 and Lhr2, from which the above aLhrs were classified as Lhr1 [24]. However, a recent report also describing a comprehensive phylogenetic analysis of the

SF2 helicases showed that the aLhrs are divided into four phylogenetically distinct groups, namely, aLhr1, aLhr2, aLhr3, and aLhr-like [25]. Archaeal Lhrs share a conserved structure containing RecA1, RecA2, and winged helix domains, but domain 4 is conserved only in aLhr1 and aLhr2. aLhr3 is characterized by a highly deteriorated domain 4 [25]. The names for aLhr1 and aLhr2 in this report were different from those described in a previous one [24].

As per the published reports, the exact functions of these candidate helicases in cells remain unclear. Thus, to expand our knowledge about the HR process in Archaea, it is important to determine which helicase is primarily involved in the branch migration process.

ssDNA-binding proteins, designated as SSB in Bacteria and Crenarchaea, or replication protein A (RPA) in Eukarya and Euryarchaea, specifically bind ssDNA without sequence-specificity via oligonucleotide-binding folds (OB-fold) [26–28]. This protein is universally distributed in cellular organisms with some exceptions [29] and plays essential roles in DNA replication, recombination, and repair [30–32]. It is known to be involved in HR in cellular life, and the functional mechanism is considered to bind to the ssDNA region of 3′-overhang DNA produced by the end resection, and protects the formation of a secondary structure of ssDNA, thereby promoting strand exchange catalyzed by the recombinase in vitro [33–36]. As another function of crenarchaeal SSB in HR, Rolfsmeier and Haseltine discussed the possibility that SSB incorporated a number of likely recombination mediators in the presynaptic step before the strand invasion of HR [37]. The role of physical and functional protein interactions involving SSBs in HR remains unresolved and argued the possibility that SSB bridged between DNA and such proteins [i.e., NurA and RadA [37,38]. As SSB of *S. solfataricus* destabilized double-stranded DNA (dsDNA)-containing lesions and possibly interacted with other proteins, Cubeddu and White hypothesized that SSB recruited proteins to damage sites via protein interaction [39]. Based on these concepts, we questioned whether SSB acts as a scaffold protein to localize proteins to HR sites via protein interaction. A previous pull-down experiment with a biotin-ssDNA saturated with SSB from the *S. solfataricus* cell extract demonstrated that some unknown proteins with a helicase-like sequence were co-purified with an ssDNA-SSB complex, suggesting that these helicase-like proteins interacted with SSB [39]. Therefore, it would be suitable to investigate the activity and function of these proteins.

In this study, based on the concepts described above, we focused on a protein Saci_0814, which is the homolog of the above helicase-like protein in *S. acidocaldarius*. We purified the Saci_0814 protein and characterized its helicase activity. In addition, we isolated the deletion mutant for the corresponding gene in the *S. acidocaldarius* genome and demonstrated a decrease in HR frequency. Our findings suggest that Saci_0814 acts as a functional branch migration helicase in this archaeon.

## 2. Results

### 2.1. Prediction of Saci_0814 as a Putative Helicase

Cubeddu and White reported candidate proteins interacting with the ssDNA-SSB complex in their study on SSB from *S. solfataricus* [39]. An SF2 helicase (SSO0394) with unknown functions was included in the candidate proteins. A homolog protein of SSO0394 was found in the *S. acidocaldarius* genome. Saci_0814 shared 54% sequence identity with SSO0394 over the entire amino acid sequence. Comprehensive phylogenetic analysis of the SF2 helicases in living organisms showed that the archaeal Lhrs were divided into four phylogenetically distinct groups, namely, aLhr1, aLhr2, aLhr3, and aLhr-like [25]. The cysteine-rich motif at the C-terminus region is a specific feature of aLhr1, and neither bacterial Lhr nor the other archaeal Lhrs possess it [25]. This region has a consensus sequence of CPKCGSRM(LIV)(AT)(AV)(LV)(KN)PW [24]. Saci_0814 has a similar sequence, <u>CTKCGSIFLTV</u>TDED (the identical sequence is underlined); therefore, we designated Saci_0814 as *Sac*aLhr1 (*alhr1* as the gene name) here, and investigated its function in vivo and in vitro.

### 2.2. Construction of the alhr1-Deleted Strain of S. acidocaldarius

To investigate the role of *Sac*aLhr1 in vivo, we constructed a deletion mutant strain of *S. acidocaldarius* for the gene encoding *Sac*aLhr1 from the parental DP-1 strain (Supplemental Figure S1A). The isolated strains were subjected to PCR using primers designed for both the inner and outer regions of *alhr1*, to confirm the deletion of the target gene from the genome. The shortened PCR product was obtained using the outer primers from the genomic DNA of the isolated strain (Supplemental Figure S1B: lanes DP-1 and 17), but no amplification product was detected by PCR with internal primers (Supplemental Figure S1C). These results indicated complete deletion of the *alhr1* gene in the genomic DNA of the strain. This strain was designated as DP-17 (Table 1).

**Table 1.** Strains or DNAs used in genetic study.

| Strains or DNAs | Relevant Characteristic (s) | Source or Reference |
|---|---|---|
| Strains | | |
| *S. acidocaldarius* | | |
| DP-1 | SK-1 [40] with Δ*phr* (Δ*pyrE* Δ*suaI* Δ*phr*) | [41] |
| DP-17 | DP-1 with Δ*alhr1* (Δ*pyrE* Δ*suaI* Δ*phr* Δ*alhr1*) | This study |
| Plasmids | | |
| placSpyrE | Plasmid DNA carrying 0.8 kb of 5′ and 3′ flanking regions of *suaI* locus at both ends of *pyrE-lacS* dual marker | [41] |
| pSAV2 | *Sulfolobus-E. coli* shuttle vector, based on pBluescript II KS (-) and pRN1, with the *SsopyrEF* maker | [40] |
| PCR products | | |
| MONSTER-alhr1 | Linear DNA carrying a 39 bp region of *alhr1* as the Tg-arm, and the 39 bp 3′ and 30 bp 5′ flanking regions of *alhr1* at both ends of *pyrE-lacS* dual marker | This study |
| pyrElacS800 | Linear DNA carrying 0.8 kb of 5′ and 3′ flanking regions of *suaI* locus at both ends of *pyrE-lacS* dual marker | [41] |

### 2.3. Estimation of HR Frequencies

To investigate the in vivo role of *Sac*aLhr1 on HR in *S. acidocaldarius*, the HR frequency of DP-17 was compared with that of the parent strain, DP-1. In this analysis, the selectable marker (*lacS-pyrE*) could only be maintained if it was integrated into the host genome by HR via a double crossover between the linear marker cassette (pyrElacS800) and the chromosome at the 5′ and 3′ homologous regions of the target locus (see Materials and Methods section). The autonomously replicating shuttle vector pSAV2 containing *pyrE* was used as a control to determine the transformation efficiency of this experiment.

The difference in transformation efficiency of DP-1 and DP-17 was not statistically significant ($p > 0.001$, two-tailed Student's *t*-test) (Supplemental Figure S2A), suggesting that the DNA uptake capacities of both strains were similar. For this reason, we considered that HR efficiency could be compared by counting transformant cells on the selection plates of the two strains. The number of transformants from DP-17 was 5-fold lower than that from DP-1 ($p = 0.00083$) (Supplemental Figure S2B). These results indicate that *Sac*aLhr1 is involved in the HR process in *S. acidocaldarius* in vivo.

### 2.4. Purification of the Recombinant Protein Encoded by alhr1

To determine the function of *Sac*aLhr1 in the HR process, biochemical characterization of this protein was necessary. Therefore, we cloned the *alhr1* gene into the expression plasmid and produced the *Sac*aLhr1 protein in *E. coli* cells. Production of the recombinant *Sac*aLhr1 protein was designed as an N-terminal His-tagged form, and it was successfully overexpressed in a soluble fraction from *E. coli* C41 (DE3) cells. The expression of *Sac*aLhr1 in the C41 (DE3) cells was crucial for increasing protein solubility because the protein was hardly produced in a soluble fraction when *E. coli* BL21 (DE3) was used as a host cell. The gel filtration profile in the last step of purification is shown in Figure 1A. The

molecular weight of the protein estimated from the elution profile was $9.7 \times 10^4$, which was approximately 106,353, as calculated from the deduced amino acid sequence of His-tagged *Sac*aLhr1, suggesting that *Sac*aLhr1 exists as a monomer in solution. The recombinant protein was purified to near homogeneity, as determined by SDS-PAGE analysis (Figure 1B). The relative molecular mass of the protein estimated from its migration position in the SDS-PAGE gel corresponded to 106,353. From a 1 L culture (7.2 g wet cells), 1.5 mg of homogeneous protein was obtained.

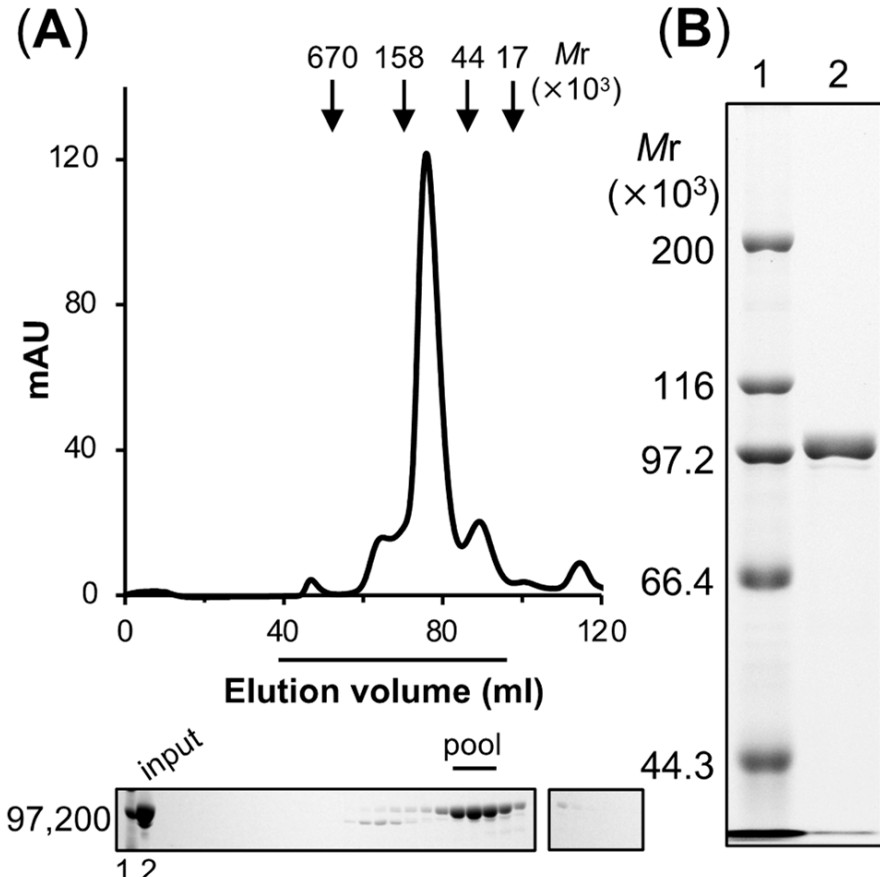

**Figure 1.** Preparation of recombinant *Sac*aLhr1 protein. (**A**) Gel filtration chromatography of *Sac*aLhr1 following His-tag purification. Arrowheads indicate the elution profile of standard marker proteins, and the numbers indicate relative molecular masses. Fractions indicated by underbar were subjected to SDS-7.5% PAGE followed by staining with Coomassie Brilliant Blue (CBB) and are shown under the chromatogram. Lanes 1 and 2 are the protein marker (*Mr*, 97,200) and input, respectively. Three fractions were pooled and used for subsequent experiments. (**B**) Protein markers (lane 1) and the purified *Sac*aLhr1 (1 μg) (lane 2) were separated using SDS-7.5% PAGE and stained with CBB. Sizes of the markers are shown to the left of the panel.

*2.5. ss/dsDNA-Dependent ATPase Activity of SacaLhr1*

The purified *Sac*aLhr1 protein was subjected to ATPase assay in the presence and absence of ssDNA or dsDNA. As shown in Figure 2, the *Sac*aLhr1 protein itself weakly hydrolyzed ATP, and this ATPase activity was clearly stimulated in the presence of increasing concentrations of ss/dsDNA. Notably, *Sac*aLhr1 is an ss/dsDNA-dependent ATPase, and ssDNA is preferable to dsDNA for its activity. The ss/dsDNA-dependent ATPase activity has also been demonstrated for *P. furiosus* Hjm, in which ATPase activity is stimulated in the presence of ss/dsDNA with the same efficiency [13]. However, the preference for ssDNA is more obvious for the ATPase activity of *M. thermautotrophicus* Hel308, which is stimulated considerably by ssDNA, but not dsDNA [12]. Different properties have also been reported for *S. solfataricus* aLhr2 [21] and *S. islandicus* PINA [16], in which ATPases

were not stimulated in the presence of ss/dsDNA. Here, the ATPase activity of *Sac*aLhr1 was clearly stimulated by both ssDNA and dsDNA.

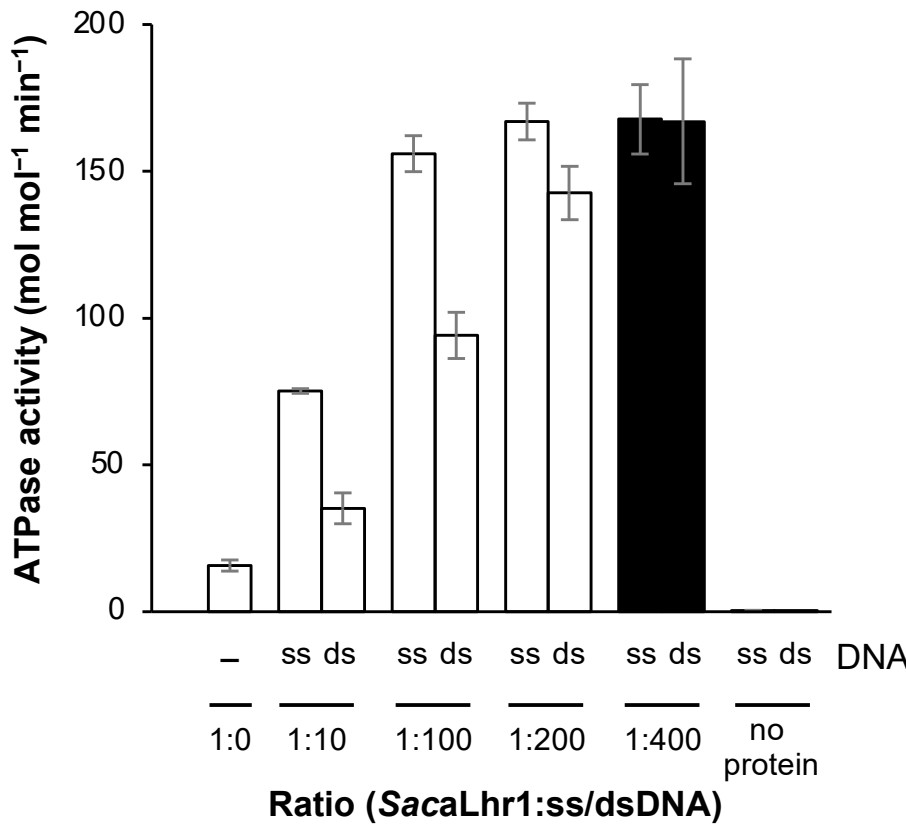

**Figure 2.** ss/dsDNA-dependent ATPase activity of *Sac*aLhr1. ATP activity refers to the amount of released Pi (mol) by a constant amount of the *Sac*aLhr1 (1 mol as a monomer) for a 1-min incubation under each reaction condition in the absence (−) or presence of ssDNA or dsDNA at a ratio of *Sac*aLhr1:ss/dsDNA = 1:0–400 and 0:400. One at the ratio means 2.25 pmol. Reactions carried out without *Sac*aLhr1, in the presence of 900 pmol ss/dsDNA (indicated as no protein) were considered as the negative control. Data are presented as mean ± standard deviation (SD), calculated from values of three independent experiments.

### 2.6. SacaLhr1 Is a 3′-to-5′ DNA Helicase

The DNA-strand unwinding activity of the *Sac*aLhr1 protein was investigated using normal duplex DNA with a single-stranded region at either the 5′- or 3′-end. The duplex DNA with 34 bp and 20 or 24 nt of the single-stranded region for 5′- and 3′-end, respectively, were used as substrates. As shown in Figure 3, *Sac*aLhr1 clearly showed unwinding activity for the DNA with 3′-protruding (Figure 3A), but not for the DNA with 5′-protruding ssDNA (Figure 3B). This result indicates that *Sac*aLhr1 unwinds the DNA strand in the 3′-to-5′ direction. It has been suggested that the 3′-to-5′ polarity of the helicase of *Sac*aLhr1 is a common function rather than a distinctive feature because it has also been reported for aLhr2 from *S. solfataricus* [21], *S. acidocaldarius* [22], and *M. thermautotrophicus* [23]; Hjm from *P. furiosus* [13]; Hel308 from *M. thermautotrophicus* [12].

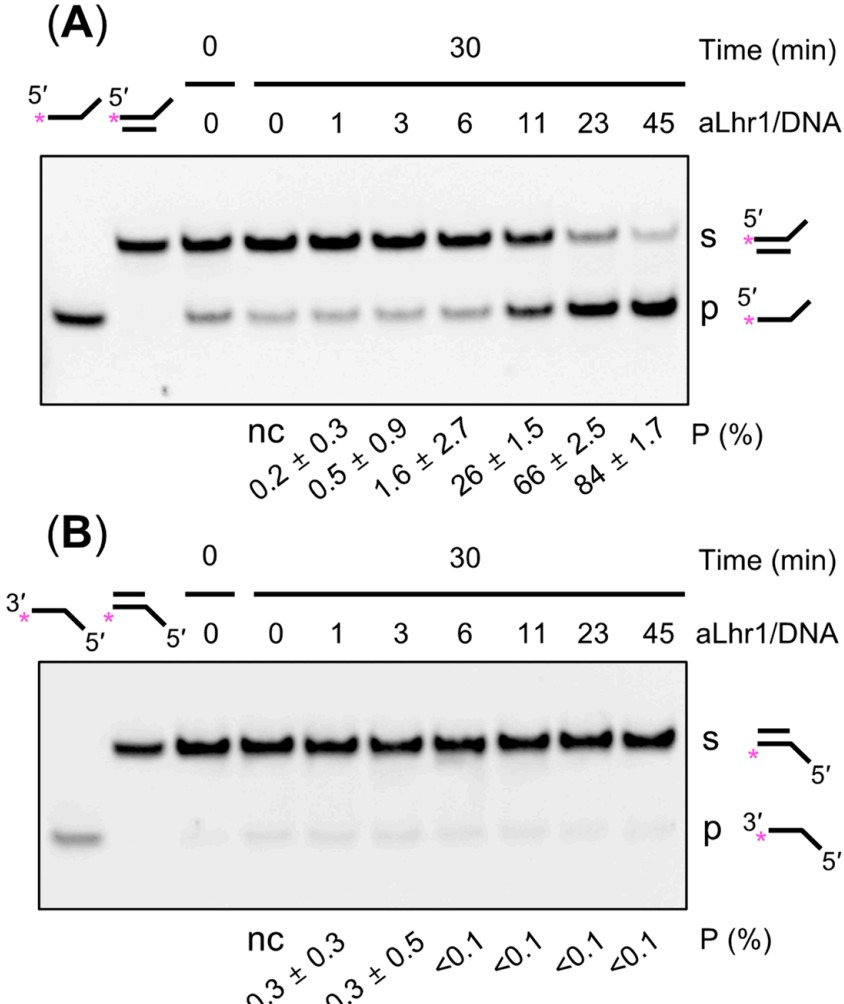

**Figure 3.** *Sac*aLhr1 is a 3′-to-5′ DNA helicase. We incubated 10 nM of 3′-protruding DNA (**A**) and a 5′-protruding DNA (**B**) with the indicated ratio of *Sac*aLhr1 and DNA (aLhr1/DNA) at 70 °C for 30 min. Asterisks in (**A,B**) indicate 5′- and 3′-Cy5 labels, respectively. The band assignments are indicated on the right side of the panels (s, substrates; p, products). Reactions were quantified by measuring the band intensities and expressed as P% (product per total DNA) and are indicated under each lane. Value from the negative control reaction without protein (nc, negative control) was subtracted from each value. Data are presented as mean ± SD, calculated from values of three independent experiments.

### 2.7. ATP- and Mg$^{2+}$-Dependent Helicase Activity of SacaLhr1

Helicase activity often requires energy from ATP hydrolysis and divalent cations as cofactors. To examine the effect of ATP and Mg$^{2+}$ on the helicase activity of *Sac*aLhr1, a helicase assay using a synthetic oligonucleotide substrate and trap DNA designed by Ogino et al. [42] was used. A splayed DNA (comprising a 34-bp duplex with a 20-nt long 5′-single-stranded tail and 36-nt long 3′-single-stranded tail) was prepared using two oligonucleotides, one of which was labeled with Cy5 at the 5′-terminus. The trap DNA can anneal with the labeled oligonucleotide after unwinding of the splayed DNA for protection from reannealing to the splayed DNA (Table 2).

**Table 2.** Substrates and trap DNAs used in this study.

| Name | Sequence (5′-3′) or Combination for Annealing |
| --- | --- |
| Labeled oligonucleotides [a] | |
| Cy5-HJ4-59 | Cy5-GACCTAGGAACCACCAGAAACACGCCACAGCCAGGAAGCCGATTGCGAGGCCGTCCTAC |
| Cy5-HJ3-54 | Cy5-TCACTCCGCATCTGCCGATTCTGGCTGTGGCGTGTTTCTGGTGGTTCCTAGGTC |
| HJ3-54mer-Cy5 | TCACTCCGCATCTGCCGATTCTGGCTGTGGCGTGTTTCTGGTGGTTCCTAGGTC-Cy5 |
| DyLight 5-HJ2S | DyLight 5-CGTTGACATCTCGCGTGCTCGGTCAATCGGCAGATGCGGAGTGAAGTTCC |
| Oligonucleotides | |
| HJ4-34 | GACCTAGGAACCACCAGAAACACGCCACAGCCAG |
| HJ4-34-R | CTGGCTGTGGCGTGTTTCTGGTGGTTCCTAGGTC |
| HJ4 | GACCTAGGAACCACCAGAAACACGCCACAGCCAGGAAGCCGATTGCGAGGCCGTCCTACCATCCTGCAGG |
| HJ1S | GGTAGGACGGCCTCGCAATCGGCTTCGACCGAGCACGCGAGATGTCAACG |
| HJ3S | GGAACTTCACTCCGCATCTGCCGATTCTGGCTGTGGCGTGTTTCTGGTGG |
| HJ4S | CCACCAGAAACACGCCACAGCCAGGAAGCCGATTGCGAGGCCGTCCTACC |
| HJ2S-RC | GGAACTTCACTCCGCATCTGCCGATTGACCGAGCACGCGAGATGTCAACG |
| HJS1-2-trap | GGAACTTCACTCCGCATCTGCCGATTGAAGCCGATTGCGAGGCCGTCCTACC |
| HJS2-3-trap | CCACCAGAAACACGCCACAGCCAGGACCGAGCACGCGAGATGTCAACG |
| DNA substrates [b] | |
| 3′-overhang | Cy5-HJ4-59/HJ4-34-R |
| 5′-overhang | HJ3-54mer-Cy5/HJ4-34 |
| Splayed DNA1 | Cy5-HJ3-54/HJ4 |
| ssDNA | DyLight 5-HJ2S |
| dsDNA | DyLight 5-HJ2S/HJ2S-RC |
| Splayed DNA2 | DyLight 5-HJ2S/HJ3S |
| Y-junction1 | HJ1S/DyLight 5-HJ2S/HJS1-2-trap |
| Y-junction2 | DyLight 5-HJ2S/HJ3S/HJS2-3-trap |
| Three-strand DNA1 | DyLight 5-HJ2S/HJ3S/HJ4S |
| Three-strand DNA2 | HJ1S/DyLight 5-HJ2S/HJ4S |
| Three-strand DNA3 | HJ1S/DyLight 5-HJ2S/HJ3S |
| HJS | HJ1S/DyLight 5-HJ2S/HJ3S/HJ4S |

[a] Substrate labeled at their 5′ or 3′ ends are denoted with Cy5 or DyLight 5. [b] Various combinations of oligonucleotides were used to generate substrates, as indicated.

The effect of ATP and Mg$^{2+}$ on the helicase activity of *Saca*Lhr1 was studied over a reaction time of 15 min. A time-dependent unwinding reaction was observed in a linear relationship (Figure 4A,B). The unwinding activity was ATP-dependent and the reaction was efficient with an ATP concentration of 0.2–2 mM (Figure 4C). A small amount of the product was detected in the absence of MgCl$_2$ (Figure 4D). However, when EDTA was added, the residual unwinding reaction was partially inhibited (Figure 4D). The helicase activity was accelerated by Mg$^{2+}$, and the reaction was efficient at a Mg$^{2+}$ concentration range of 2.5–15 mM (Figure 4D). These results indicate that *Saca*Lhr1 is an ATP- and Mg$^{2+}$-dependent helicase. This condition (1 mM ATP and 5 mM MgCl$_2$) was used for subsequent assays.

*2.8. Unwinding Activity SacaLhr1 on Branched DNA*

To investigate whether *Saca*Lhr1 has branch migration activity, dissociation activity against a synthetic HJ (HJS: 50-nt long oligonucleotides forming a four-way junction with no homologous region) (Table 2) by *Saca*Lhr1 was characterized (Figure 5). If the protein had a branch migration activity for the HJ DNA, a half junction (Y-structure or splayed DNA) could be formed from HJS. However, *Saca*Lhr1 unwound HJS, producing ssDNA as the main product (Figure 5). This unwinding property of *Saca*Lhr1 is similar to that of aLhr2 from *S. acidocaldarius* [22] but not those of other candidates for branch migration helicase, for which ssDNA and a half junction were detected as products at similar levels [11,16,23]. *Saca*Lhr1 also unwound a synthetic three-way junction (Y-junction2), but not normal duplex DNA, producing ssDNA (Figure 5). At this point, it is not clear whether *Saca*Lhr1 has

branch migration activity for HJ-structured DNA. However, this enzyme has a strand unwinding activity, specifically for Y-junction2 and HJS, which contain a branch point that produces ssDNA.

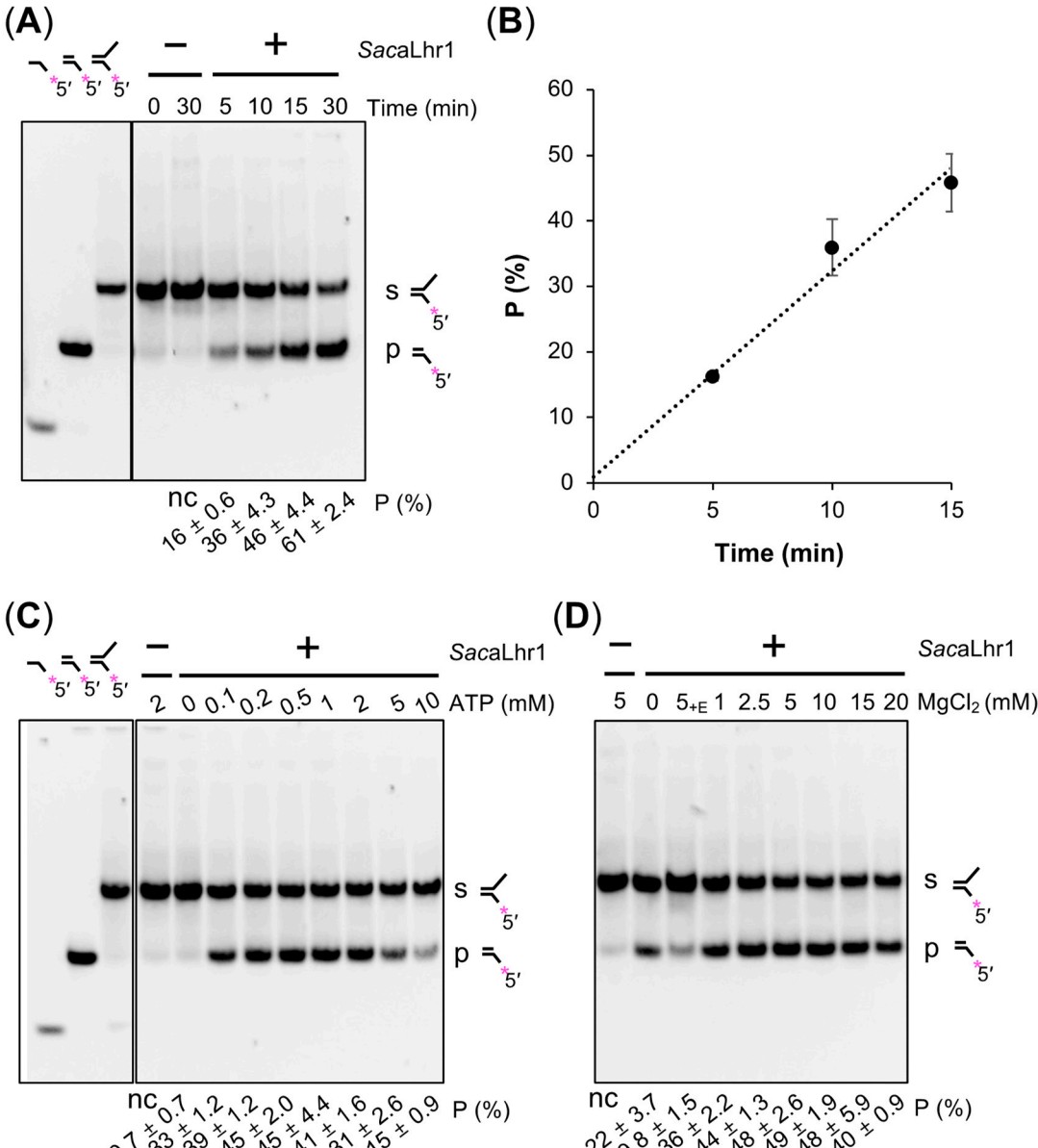

**Figure 4.** ATP- and $Mg^{2+}$-dependent helicase activity of *Saca*Lhr1. (**A**) Time-course analysis of helicase activity. Splayed DNA1 (10 nM) was incubated with (+) or without (−) *Saca*Lhr1 (225 nM) at 60 °C for 0–30 min. (**B**) Reactions in (**A**) were quantified and plotted. SD was calculated from three independent experiments. ATP (**C**) and $Mg^{2+}$ (**D**) dependences of the helicase activity were examined. Reactions were performed at 60 °C for 15 min in the presence of 5 mM $MgCl_2$ with indicated concentrations of ATP in (**C**) and 2 mM ATP with indicated concentrations of $MgCl_2$ in (**D**). +E indicates reaction containing 0.1 M EDTA. Asterisks represent 5′-Cy5 labels. The band assignments are indicated on the right side of the panels, s, substrates; p, products. Reactions were quantified by measuring the band intensities and expressed as P% (product per total DNA) and are indicated under each lane. Value from the negative control reaction without protein (nc, negative control) was subtracted from each value. Data are presented as mean ± SD, calculated from values of three independent experiments.

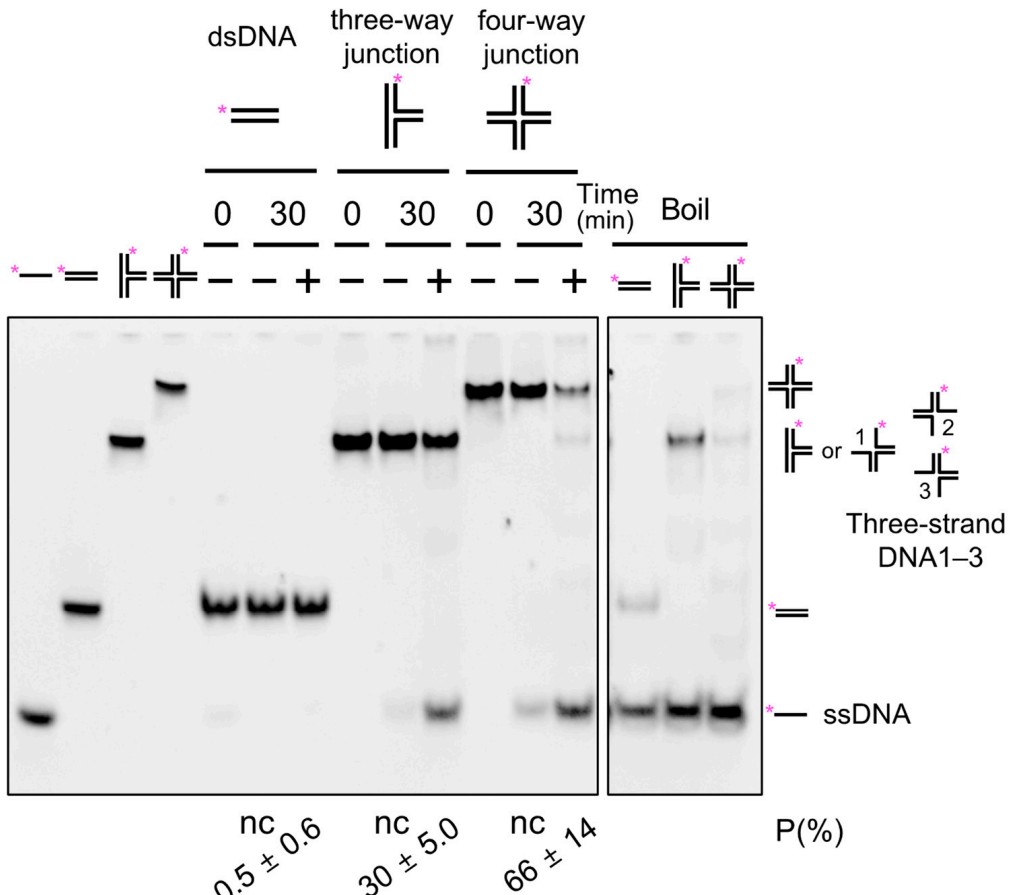

**Figure 5.** Helicase activity of *Sac*aLhr1 for branched DNA molecules. dsDNA, three-way junction (Y-junction2), and four-way junction (HJS) (10 nM) were incubated with (+) or without (−) *Sac*aLhr1 (123 nM) at 70 °C for 30 min. Reactions were performed without trap DNA. Asterisks represent 5′-DyLight 5 labels. "Boil" indicates boiling of the substrate at 98 °C for 3 min followed by cooling on ice water. Detected bands were assigned using the controls of each shaped DNA on the right side of the panel. Reactions were quantified by measuring the band intensities and expressed as P% (product per total DNA) and are indicated under each lane. Value from each negative control reaction without protein (nc, negative control) was subtracted from each value. Data are presented as mean ± SD, calculated from values of three independent experiments.

### 2.9. Binding Properties of SacaLhr1 to DNA with or without a Branch Point

To determine whether *Sac*aLhr1 specifically recognizes the branch point of DNA, the binding properties of *Sac*aLhr1 to DNA substrates with or without the branch point (ssDNA, dsDNA, splayed DNA2, Y-junction2, and HJS) were examined. The binding preference of *Sac*aLhr1 was not significantly different among the DNA substrates (Figure 6). Multiple shifted bands were observed for all the DNA substrates used for the EMSA. It was not easy to assign each shifted band at this stage, but it was clear that *Sac*aLhr1 preferably bound the DNA molecules with a single-stranded region (ssDNA and splayed DNA2) (Figure 6A–C). *Sac*aLhr1 bound to HJS, Y-junction2, and dsDNA with about the same efficiency (Figure 6B,D,E). The binding preference of *Sac*aLhr1 to ssDNA, compared with dsDNA, was consistent with the result of its DNA-dependent ATPase activity, wherein ATPase activity was more stimulated by ssDNA than dsDNA (Figure 2). No specific binding to a branch point by *Sac*aLhr1 was demonstrated. The binding properties of *Sac*aLhr1 are similar to those of Hjm from *P. furiosus* [13]. However, no clear difference in the binding properties of *Sac*aLhr1 to synthetic HJ and dsDNA were observed. This lack of binding preference for *Sac*aLhr1 is distinct from that of aLhr2 from *S. acidocaldarius*, Hjm

from *S. tokodaii*, Hel308 from *M. thermautotrophicus*, and PINA from *S. islandicus*. These helicases preferably bind to a synthetic HJ over dsDNA [12,15,16,22].

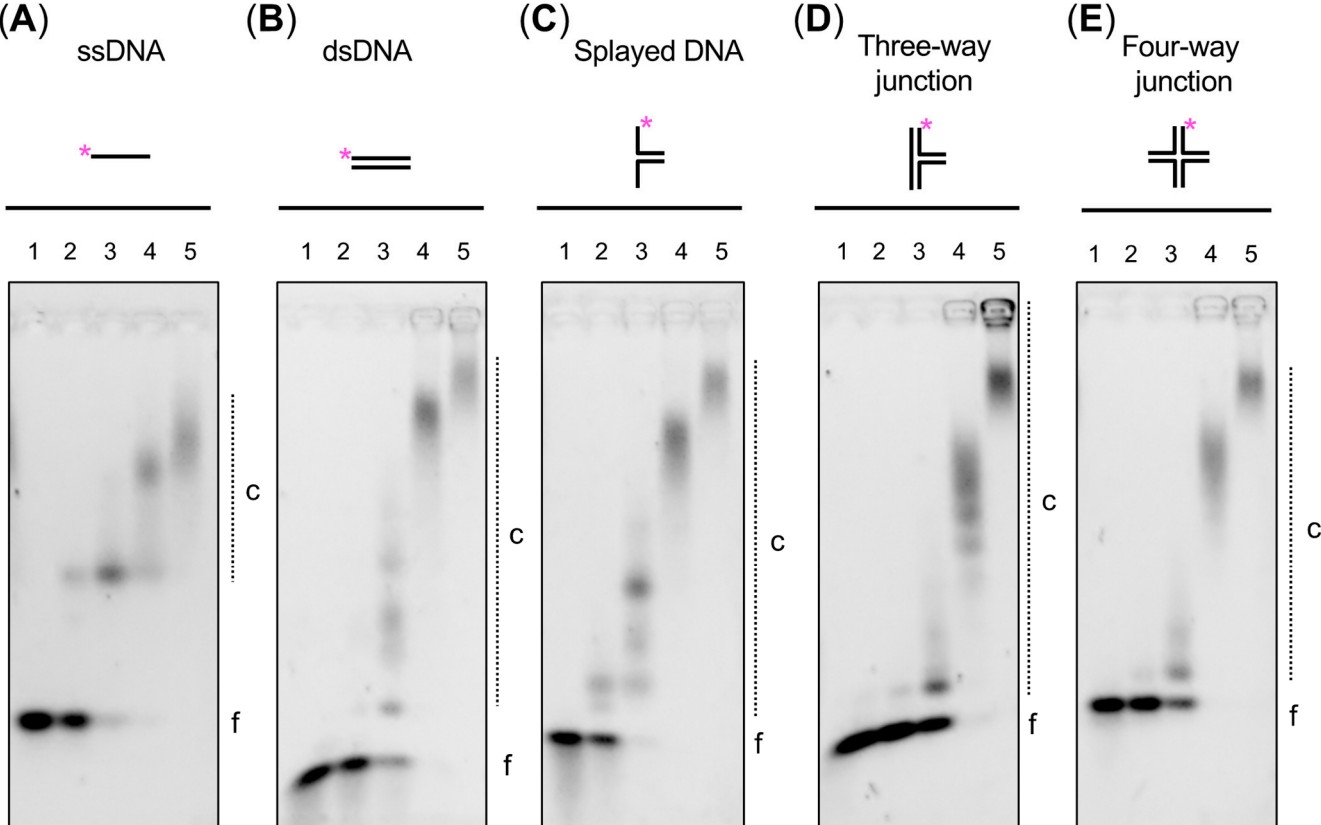

**Figure 6.** DNA-binding properties of *Sac*aLhr1. DNA molecules (10 nM) with various structures (ssDNA (**A**), dsDNA (**B**), splayed DNA2 (**C**), Y-junction2 (**D**), and HJS (**E**)) were incubated with increasing concentrations of *Sac*aLhr1 (lane 1, no protein; 2, 60 nM; 3, 110 nM; 4, 230 nM; 5, 450 nM) at 25 °C for 10 min. Asterisks in (**A–E**) represent 5′-DyLight 5 labels. Products were separated by 1% or 2% agarose gel in 0.1× TBE. Band assignments are indicated on the side of the panels (c, protein-DNA complexes; f, free DNA). Experiments were repeated thrice, and representative gel images are shown.

### 2.10. Branch Migration Activity of SacaLhr1 against a Synthetic HJ

*Sac*aLhr1 unwound the synthetic HJ, HJS, producing ssDNA (Figure 5). Regarding the unwinding reaction, two processes were considered as possible mechanisms: (i) *Sac*aLhr1 binding a single-stranded region generated by thermal denaturation at high temperature followed by direct unwinding of HJS in the 3′-to-5′ direction from the produced ssDNA. (ii) The inability of *Sac*aLhr1 to unwind the duplex DNA without a 3′-single-stranded tail, as shown above, implied that it was unlikely that *Sac*aLhr1 directly unwound HJS to produce ssDNA. As shown in Figure 7A, *Sac*aLhr1 possibly first dissociates HJS to the half junctions, using its branch migration activity, and then unwinds the half junctions to the ssDNA in the 3′-to-5′ direction. Thus, we assessed whether *Sac*aLhr1 dissociated HJS to half junctions using its branch migration activity.

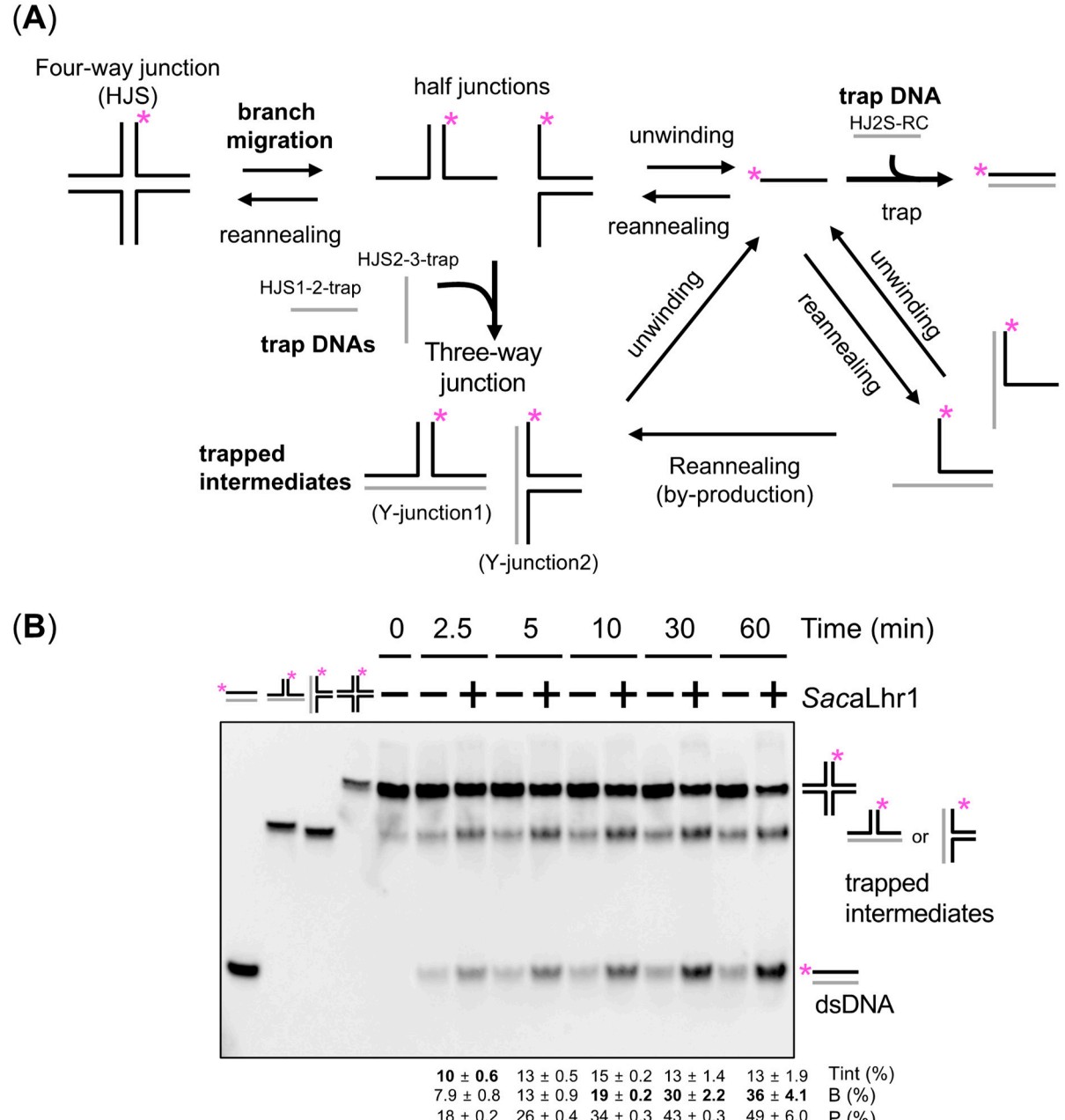

**Figure 7.** Branch migration activity of *Saca*Lhr1 for synthetic HJ. (**A**) Schematic drawing of the possible unwinding and reannealing processes of substrate DNA. Gray lines and asterisks indicate the trap DNA molecules and 5′-DyLight 5 labels, respectively. (**B**) Synthetic HJ (10 nM) were incubated with (+) or without (−) *Saca*Lhr1 (225 nM) at 70 °C for 0–60 min. Bands were assigned and indicated on the right side of the panel. Amounts of trapped intermediates (Tint%) (trapped intermediates per total DNA), dsDNA (B%) (dsDNA per total DNA), and product (P%) (trapped intermediates and dsDNA per total DNA) were quantified using band intensities obtained via scanning the gel image and indicated under the panel. For each band, intensity from the control reaction without protein was subtracted. Experiments were repeated thrice, and representative gel images are shown.

To evaluate the formation of half junctions, an assay to trap the half junctions was developed (Figure 7A). Under these conditions, three trap DNA molecules were designed. Two of the trap DNA molecules (HJS1-2-trap and HJS2-3-trap) could hybridize to each of the two half-junctions produced from HJS (Table 2), thus producing the trapped intermediates Y-junction1 and Y-junction2 (Figure 7A). The third trap DNA (HJ2S-RC) comprised a

complementary strand of the 5′-DyLight 5 labeled ssDNA (one of the four DNA strands consisting of HJS) to prevent reannealing the labeled ssDNA with the other trap DNA (HJS1-2-trap and HJS2-3-trap) and the unwound DNA (HJ1S and HJ3S) from half junctions (Figure 7A).

The unwinding process of HJS, catalyzed by *Sac*aLhr1, was examined in a time-dependent manner in the presence of the three trap DNA molecules. As shown in Figure 7B, the number of trapped half junctions (Tint%) was higher than that of the trapped ssDNA (B%), but B% increased with longer reaction times. Quantification of each band showed that Tint% was higher than that of B% at 2.5 min. However, B% increased with increasing reaction times, and the difference values was reversed after 10 min. Tint% without protein was constant throughout the reaction. These results suggest that *Sac*aLhr1 promotes the branch migration reaction for synthetic HJ.

## 3. Discussion

Homologous recombination plays a very important role in maintaining the genome stability of HA. However, its molecular mechanism in Archaea is not fully elucidated. From the many studies on each step of HR in Archaea, Hjm/Hel308 [11,12,15], PINA [16], and aLhr2 [23] have been characterized in vitro as candidate branch migration helicases. However, it remains unclear whether these candidates are essential for branch migration to process the recombination intermediate in Archaea in vivo. In this study, we noted that the *alhr1*-deleted strain of *S. acidocaldarius* exhibited a defect in HR frequency (5-fold decrease) when compared with the parental strain (Supplemental Figure S2). A similar decrease in HR frequency has been demonstrated by the deletion of the genes encoding recombinase mediator proteins (Rad22 and Rhp57) in *Schizosaccharomyces pombe* (2- and 5-fold decrease) [43,44]. However, the decrease in HR frequency in the *alhr1*-deleted strain was lower than that in the recombinase-deleted eukaryotic and archaeal strains (15- or 31-fold decrease in *S. pombe* and >500–600-fold decrease in *Haloferax volcanii*) [43–45], but plasmid integration by single-crossover recombination was utilized to estimate HR frequency in previous studies. In our genetic assay, we considered that the selectable marker *lacS-pyrE* was inserted in the genomic locus by HR via double-crossover consisting of end resection of the linear marker cassette, strand exchange, formation of double HJs, branch migration, and HJ resolution. Thus, the decrease in HR frequency by the deletion of the *alhr1* gene (Supplemental Figure S2) suggests that *Sac*aLhr1 is directly involved in the HR process. To investigate whether *Sac*aLhr1 has branch migration activity in vitro, we purified *Sac*aLhr1 and characterized it as a monomer in solution and a ss/dsDNA-dependent ATPase (Figures 1 and 2). *Sac*aLhr1 unwound the DNA duplex with 3′-overhang in the 3′-to-5′ direction, and dissociated the synthetic HJ in vitro (Figures 3 and 7). Because *Sac*aLhr1 has branch migration activity (Figure 7), we suggest that *Sac*aLhr1 promotes branch migration of double HJs formed at both homologous ends of the linear marker cassette and extends heteroduplex regions to increase the frequency of crossover recombination, as noted in our genetic assay (Supplemental Figure S2).

As other candidate branch migration helicases, namely, Hjm from *P. furiosus* [11], aLhr2 from *M. thermautotrophicus* [23], and PINA from *S. islandicus* [16], also dissociate a synthetic HJ in vitro, we considered *Sac*aLhr1 as another potential branch migration helicase. Although *Sac*aLhr1 does not specifically bind to the synthetic HJ (Figure 6), it may bind to the dsDNA region of the synthetic HJ and promote branch migration via translocation on dsDNA driven by dsDNA-dependent ATPase activity (Figures 2 and 7). Furthermore, because the dsDNA-dependent ATPase activity of *Sac*aLhr1 is considered to be a distinctive feature in comparison with the candidates for branch migration helicase from other Archaea (Figure 2) [12,16,21], we speculated that Hjm from *P. furiosus* promoted branch migration via the same mechanism because Hjm also has dsDNA-dependent ATPase activity and binds to a synthetic HJ and dsDNA with mostly the same efficiency [13]. In addition, ssDNA was detected as a product in the helicase reaction by the other candidates for branch migration helicase [11,16,23], suggesting that *Sac*aLhr1 also processed the

synthetic HJ to ssDNA (Figure 5) by the same process of branch migration followed by a 3′-to-5′ helicase reaction. Based on our genetic and biochemical analyses, we suggested that *Sac*aLhr1 was involved in the HR process and may act as a branch migration helicase in this archaeon in vivo.

Regarding the decrease in HR frequency in our genetic assay (Supplemental Figure S2), we could not eliminate the possibility that *Sac*aLhr1 stimulates end resection of the linear marker cassette for HR. Similarly, BLM helicase, a member of the RecQ family in humans, stimulates dsDNA resection by EXO1 (5′-to-3′ dsDNA exonuclease) or DNA2 (helicase/ssDNA endonuclease) in vitro, which are considered to be components of DNA end resection in humans [46,47]. As a functional counterpart in Archaea, DNA helicase HerA from *S. solfataricus* stimulates (restores) dsDNA resection via the partner exo/endonuclease NurA in the presence of ATP in vitro [48]. It is likely that nucleases and helicases play important roles in the end resection step. De Falco et al. [49] demonstrated that aLhr2 from *S. solfataricus* inhibited nuclease activity of NurA on ss/dsDNA, but not nicking activity on a supercoiled plasmid DNA, in the presence of HerA and in the absence of ATP in vitro. However, its biological significance is not clear because the deletion of the aLhr2-encoding gene in *S. acidocaldarius* does not affect HR frequency in vivo [22]. It appears that no biochemical studies to date have shown whether aLhr1 physically and functionally interacts with any nucleases; thus, it was difficult to assess the validity of this possibility at this stage. Further research is required to examine whether *Sac*aLhr1 is involved in the end resection.

In terms of the role of *Sac*aLhr1 in the HR process, in addition to the branch migration activity (Figure 7), we found that *Sac*aLhr1 preferentially binds to the ssDNA (Figure 6) and has 3′-to-5′ helicase activity (Figure 3). The cellular significance of its 3′-to-5′ helicase activity is still unclear; a possible interpretation could be that the helicase activity of *Sac*aLhr1 is involved in the DNA repair process. Previous phylogenetic analysis of SF2 helicases in Archaea suggests that Hjm/Hel308, aLhr1, and aLhr2 seem to have evolved from a common ancestral helicase and share similar functions [24]. In fact, Hjm/Hel308 and aLhr2 have both branch migration activity and 3′-to-5′ helicase activity [12,13,22,23]. In addition, expression of Hjm from *P. furiosus*, or Hel308 or aLhr2 from *M. thermautotrophicus*, was observed in *E. coli dnaE486 ΔrecQ* cells mimicking the RecQ-like growth defect phenotype [12,13,23,50]. It has been proposed that RecQ unwinds a stalled replication fork (i.e., fork DNA with a gap on the leading strand) with a 3′-to-5′ polarity by two-step reactions [50]. Thus, previous genetic results suggest that Hjm/Hel308 and aLhr2 are involved in the repair of stalled replication forks. Based on these observations, the 3′-to-5′ helicase activity of *Sac*aLhr1 may also be involved in the repair of a stalled replication fork. We did not investigate how *Sac*aLhr1 works for a fork-structured DNA in this study, further biochemical studies are required to elucidate the stalled fork repair process.

Lhr was originally discovered in *E. coli*, which was followed by continuous biochemical characterization of the homologs from *Mycobacterium* and *Pseudomonas*. These studies indicated that bacterial Lhrs are ssDNA-dependent ATPases and 3′-to-5′ DNA translocases with unwinding properties both on RNA:DNA and DNA:DNA hybrids [19,20,51]. The capacity to unwind the RNA:DNA hybrid was also reported for aLhr2 from *M. thermautotrophicus* and *Thermococcus barophilus* [23,25]. The aLhr1 (800–900 aa) is shorter than those from bacteria (1400–1700 aa), which are now classified as bLhr-HTH [25]. The extensional C-terminal domain of bLhr-HTH is required for the formation of a homo-oligomeric quaternary structure [52]. The lack of this C-terminal region in *Sac*aLhr1 is consistent with its monomeric structure in solution (Figure 1A). aLhr1 was found in some DPANN, Asgard, and almost all TACK genomes, in addition to most of the euryarchaeal genomes, except for those of *Methanopyraceae*, *Methanococcales*, and *Methanobacteriales* [25]. Based on the nearly ubiquitous distribution of aLhr1 homologs in Archaea, we suggest that aLhr1 is important for HR in Archaea. Moreover, *Methanopyraceae*, *Methanococcales*, and *Methanobacteriales* may have another counterpart to compensate for the aLhr1 function. Genomic analysis showed that the *alhr2*, but not *alhr1*, gene of the *Sulfolobales* is located near the genes encoding the

UV-inducible pili [22,25], suggesting that aLhr1 and aLhr2 play distinct roles in *Sulfolobales* in vivo. Song et al. [53] indicated that the *alhr1* (SiRe1605)-deleted strain of *S. islandicus* exhibited sensitivity to MMS, suggesting that aLhr1 is involved in MMS-induced DNA repair for the alkylation damages. Wolferen et al. [22] argued that aLhr2 of *S. acidocaldarius* processes HJ created after UV irradiation because the *alhr2*-deleted strain exhibited sensitivity to UV irradiation without showing defects in HR frequency. These results suggest that aLhr1 and aLhr2 are involved in DNA repair in *Sulfolobales*. Currently, it is not clear whether aLhr2 is involved in the HR process in *S. acidocaldarius.* Genetic analysis of *alhr1* and *alhr2* double knockout strains will reveal whether the functions of aLhr1 and aLhr2 in HR overlap.

The structure of aLhr1 has been partially predicted and it is known to have a C-terminal domain containing a Zn-finger-like motif and helix-turn-helix motif (784-911 aa), which are unique to aLhr1 and not present in other Lhr family proteins. The crystal structure of the *Mycobacterium smegmatis* Lhr core domain (1–856 aa) in complex with AMPPNP, $Mg^{2+}$, and ssDNA has been elucidated [54]. This domain possesses the helicase and translocase activities, and its structure is conserved in the aLhr2 and Hel308/Hjm from *M. thermoautotrophicus* [23,55]. These proteins primarily participate in stalled replication fork repair. Since the 3D structure of aLhr1 has not yet been determined, it is difficult to precisely predict the structure-function relationship. However, we speculate that, the unique structural features of the C-terminal domain of aLhr1 might be responsible for its affinity for specific DNA structures. Due to the differences in structure of aLhr1, its affinity towards preferred DNA substrates also varies from other Lhr family helicases. As this study focused on the characterization of *Sac*aLhr1, we did not analyze the SSB protein. However, we think it is important to study on the relationship between *Sac*aLhr1 and SSB are necessary. Future analysis will include the purification of SSB protein from *S. acidocaldarius*, and physical and functional analysis with *Sac*aLhr1. These experiments will elucidate whether SSB interacts with *Sac*aLhr1 in vitro and whether both proteins colocalize in vivo, and will provide new insights into the molecular mechanism of HR processing by *Sac*aLhr1.

## 4. Materials and Methods

### 4.1. Strains and Growth Conditions

The strains used in this study are listed in Table 1. The *S. acidocaldarius* strain DP-1 (Δ*pyrE* Δ*suaI* Δ*phr*), in which pyrimidine auxotrophy can be complemented by the selectable marker *pyrE* [41], was used for the targeted disruption of *alhr1*. Cultivation and transformation of *S. acidocaldarius* DP-1 and its derivatives were performed as reported previously [41].

### 4.2. Construction of Gene-Deleted Strain

The plasmids and PCR products used in this study are listed in Table 1. A multiple gene knockout system with one-step PCR (MONSTER) [41] was utilized to prepare the cassette (MONSTER-alhr1) and for construction of the *alhr1*-deleted strain. In brief, MONSTER-alhr1 was amplified from placSpyrE via PCR using primers MONSTER-alhr1-F (5′-GAAAGAAAGTTTTTAAGAACACGCAAGTACTGGCAGTAAGTTTTTCTCTATATCA ATCTC-3′), comprising a 39-bp target gene (Tg), arm, and a sequence annealed with the *lacS* marker gene, and MONSTER-alhr1-R (5′-ACCTAACCTAAAAAACAAGAGTTGTATAAA AATATAAACAAACTACATAATAACCTTATTTCTGAGACTCTCCTAGATCTAAAACTAA AG-3′), comprising a 39-bp 3′-flanking region and a 30-bp 5′-flanking region of the *alhr1 gene* and a sequence that anneals with the *pyrE*-maker gene. The amplified DNA fragment was used for transformation.

The transformation protocol for *S. acidocaldarius* has been previously described in detail [41]. To disrupt the *alhr1* gene, 2 μg of MONSTER-alhr1 was electroporated into DP-1 cells harvested at the late-log phase. After electroporation, the cells were spread on a selective uracil-free xylose and tryptone (XT) plate medium. After cultivation for 6 d at

75 °C, the transformants grown on the plate were treated with X-gal solution (10 mg/mL) and were further cultivated at 75 °C for 1 d. The transformant forming blue colony was isolated, and the genomic DNA was analyzed via PCR using outer primer sets F1/R1 (5′-AATATCCAGTATTAACTAAATAC-3′ and 5′-GTTACTAGAACAAAATTACTATC-3′) to detect the insertion of MONSTER-alhr1. This intermediate strain was named DP-17 Int (*pyrE*⁺ *lacS*⁺) and was used for pop-out recombination. After cultivation on XTUF plate media followed by X-gal visualization, white colonies were selected and analyzed by colony PCR using outer primer sets. The *alhr1* gene disruptant (Δ*pyrE* Δ*suaI* Δ*phr* Δ*alhr1*) was designated as DP-17. The deleted region was confirmed by sequencing using outer primer sets. To confirm the complete deletion of the target gene from the genome, PCR analysis was performed using internal primer sets F2/R2 (5′-ATGATAAGTGTTTCTCAGACATTTATC-3′ and 5′-TTACTGCCAGTACTTGCGTG-3′) that were annealed within the *alhr1* gene.

### 4.3. Estimation of HR Frequencies

Estimation of HR frequencies was carried out as reported previously [41] with slight modifications. A linear marker cassette, pyrElacS800, harbored approximately 800-bp 5′- and 3′-flanking regions of the *suaI* locus attached to both ends of the *pyrE-lacS* marker (Table 1). Transformation of competent cells with pyrElacS800 was performed via electroporation. The cells were spread on XT plate media and incubated at 75 °C for 7–8 d. Transformants were selected using *pyrE*⁺ positive selection. The transformants that presented a blue color by X-gal treatment were assessed as HR-positive. As a control experiment, an autonomously replicating plasmid pSAV2 containing the *pyrE* selectable marker [40] was used for transformation to calculate the transformation efficiency of each competent cell. The HR frequency and transformation efficiency were defined as the number of positive transformants per 1 μg DNA for both assays. Statistical significance was examined using two-tailed Student's *t*-test analysis. Differences were considered statistically significant at $p < 0.001$.

### 4.4. Cloning of Gene Encoding SacaLhr1 from S. acidocaldarius

The gene encoding *Sac*aLhr1 was amplified via PCR using *S. acidocaldarius* DSM639 genomic DNA with the forward primer pET-SacaLhr1-NdeI-F (5′-CGCG<u>CATATG</u>ATAAGTGTTTCTCAGACATTTATC-3′; the *Nde*I restriction site is underlined) and the reverse primer pET-SacaLhr1-NotI-R (5′-GGGG<u>GCGGCCGC</u>TCACTGCCAGTACTTGCGTG-3′; the *Not*I restriction site is underlined) with PfuDNA polymerase. The amplified fragment was digested with *Nde*I and *Not*I and ligated to the corresponding sites of the pET22-28TEV expression vector, which is a modified plasmid of pET28a, wherein the thrombin recognition site and the kanamycin-resistant gene are replaced with a TEV protease recognition site and an ampicillin resistance gene, respectively. The resultant plasmid was designated as pET22-28TEV-SacaLhr1 expressing N-terminal His-tagged *Sac*aLhr1. The nucleotide sequence of the inserted DNA was confirmed via the dideoxy-sequencing method.

### 4.5. Preparation of the Recombinant SacaLhr1 Protein

*E. coli* C41 (DE3) (Lucigen) is a mutant strain of BL21 (DE3), in which there is a lower expression level of T7 RNA polymerase [56], and this strain is useful for the production of proteins that are toxic to *E. coli* [57]. *E. coli* C41 (DE3) was transformed with the pACYC-based plasmid containing extra copies of *argU*, *ileY*, and *leuW* tRNA, in which the chloramphenicol-resistant gene was replaced with the streptomycin-resistant gene, designated as *E. coli* C41 CodonPlus (DE3)-RIL. This *E. coli* cell harboring pET22-28TEV-SacaLhr1 were cultured in LB medium containing 50 mg/L ampicillin and 50 mg/L streptomycin at 37 °C until an $OD_{600}$ value of 0.3 was obtained. IPTG was added to a final concentration of 0.5 mM, and the cells were further cultured overnight at 25 °C. The cells were collected, resuspended in 25 mL buffer A (50 mM Tris-HCl, pH 8.0, and 1 M NaCl), and sonicated. The soluble cell extract was heated at 70 °C for 20 min. The heat-resistant fraction was slowly mixed with Ni-NTA agarose (QIAGEN) at 4 °C for 1 h. The resin was washed with

buffer A and buffer B (50 mM Tris-HCl, pH 8.0, and 0.15 M NaCl), followed by elution of the His-tagged *Sac*aLhr1 with the elution buffer containing 50 mM Tris-HCl, pH 8.0, 0.5 M NaCl, 0,3 M imidazole, and 10% glycerol. The fractions containing *Sac*aLhr1 were pooled and subjected to a Hiload 16/60 Superdex 200 pg column (GE Healthcare) with a gel filtration buffer containing 50 mM Tris-HCl, pH 8.0, 0.5 M NaCl, 0.1 mM EDTA, and 10% glycerol. Standard marker proteins, including thyroglobulin (670,000), $\gamma$-globulin (158,000), ovalbumin (44,000), and myoglobin (17,000), were subjected to gel filtration as controls. The protein concentration was calculated by measuring the absorbance at 280 nm using a theoretical molar extinction coefficient of 91,680 $M^{-1}$ $cm^{-1}$.

### 4.6. ATPase Assay

The purified *Sac*aLhr1 protein (225 nM) was incubated with 2 mM ATP in a 30-$\mu$L reaction solution (25 mM Tris-HCl, pH 8.0, 5 mM $MgCl_2$, 50 mM NaCl, and 1% glycerol) at 60 °C for 3–12 min in the presence or absence of 2.25–90 $\mu$M (mononucleotide) of M13mp18ssDNA and 2.25–90 $\mu$M (base pairs) of pUC18. The amounts of Pi in the aliquots were analyzed using an EnzChek Phosphate Assay kit (Invitrogen) according to the manufacturer's protocol. The mean and SD were calculated based on three independent experiments.

### 4.7. Preparation of DNA Substrates

Oligonucleotides used as DNA substrates were obtained from Hokkaido System Science and Sigma-Aldrich and are listed in Table 2. The combinations of oligonucleotides used for the substrates are shown in Table 2. Annealing of the appropriate oligonucleotides was performed in 20 mM Tris-HCl (pH 8.0) and 1 mM $MgCl_2$. The mixtures of the oligonucleotides were heated at 98 °C for 3 min, then gradually cooled to room temperature using a thermal cycler (GeneAtlas G, ASTEC, Kasuya-gun, Fukuoka, Japan), and stored at 4 °C for subsequent experiments.

### 4.8. DNA Unwinding Assay

DNA unwinding reactions were generally performed in mixtures containing 10 nM substrate and various concentrations of *Sac*aLhr1 proteins. The reaction was quenched by the addition of 25 mM EDTA and 0.1% SDS. The samples were analyzed by 0.1% SDS-10% PAGE in 0.5× TBE buffer (44 mM Tris, 44 mM boric acid and 1.3 mM EDTA, pH 8.3). The products were visualized with FUSION SOLO.7S.EDGE (Vilber-Lourmat) and quantified using the application software. The SD was calculated from three independent experiments. To determine the unwinding direction of *Sac*aLhr1, 10 nM 3′-overhang DNA and 5′-overhang DNA with 15 nM trap DNA (HJ4-34 and H4-34-R, respectively) were incubated with various concentrations of *Sac*aLhr1 (0, 14, 28, 56, 113, 225, and 450 nM as a monomer) in a 15 $\mu$L reaction mixture (25 mM Tris-HCl, pH 8.0, 1 mM DTT, 2 mM ATP, 5 mM $MgCl_2$, 100 mM NaCl, and 2% glycerol) at 70 °C for 30 min. For analyses of ATP and $Mg^{2+}$ dependence, 10 nM splayed DNA1 and 30 nM trap DNA (HJ4-34) were incubated with 225 nM *Sac*aLhr1 in a 15 $\mu$L reaction mixture (25 mM Tris-HCl, pH 8.0, 1 mM DTT, 50 mM NaCl, and 1% glycerol) in the presence and absence of various concentrations of ATP and $MgCl_2$ at 60 °C for the indicated time. To analyze the helicase activity of *Sac*aLhr1 against branch DNA, 10 nM dsDNA, Y-junction2, and HJS were incubated with 123 nM *Sac*aLhr1 in a 15 $\mu$L reaction mixture (25 mM Tris-HCl, pH 8.0, 1 mM DTT, 1 mM ATP, 5 mM $MgCl_2$, 27 mM NaCl, 5% glycerol, and 0.01% NP-40) at 70 °C for 30 min. To detect intermediate half junctions, 10 nM HJS with 30 nM HJS1-2-trap and HJS2-3-trap were incubated with 225 nM *Sac*aLhr1 in a 15 $\mu$L reaction mixture (25 mM Tris-HCl, pH 8.0, 1 mM DTT, 1 mM ATP, 5 mM $MgCl_2$, 50 mM NaCl, 1% glycerol, 5% PEG4000, and 0.01% NP-40) at 70 °C for 30 min. To trap ssDNA, 10 nM HJ2S-RC was used.

### 4.9. Electrophoretic Mobility Shift Assay (EMSA)

Various concentrations (0, 56, 113, 225, and 450 nM) of *Sac*aLhr1 were incubated with 10 nM ssDNA (5-DyLight 5-HJ2S), dsDNA, splayed DNA2, Y-junction2, and HJS in a 15 $\mu$L

reaction mixture (25 mM Tris-HCl, pH 8.0, 1 mM DTT, 5 mM MgCl$_2$, 100 mM NaCl, 2% glycerol) at 25 °C for 10 min, followed by the addition of EMSA buffer (25 mM EDTA, 2.5% Ficoll, and 0.1% Orange G). The protein-DNA complexes were fractionated using 1% (for ssDNA) or 2% agarose gel in 0.1× TBE buffer (9 mM Tris, 9 mM boric acid, and 0.3 mM EDTA, pH 8.3) and visualized using an image analyzer (FUSION SOLO, Vilber-Lourma).

**Supplementary Materials:** The following are available online at https://www.mdpi.com/article/10.3390/catal12010034/s1, Figure S1: Construction of *alhr1*-deleted strain using MONSTER, Figure S2: Transformation efficiency and HR frequency.

**Author Contributions:** Y.I., S.I. and S.S. conceived the project, and each author made substantial contributions to the conception or design of the work, acquisition, analysis, and interpretation of data; S.S. prepared the proteins with the help of T.Y. and performed biochemical analyses with the help of S.I., T.N. and S.M.; Genetic analyses were performed by S.S. with the help of N.K.; S.S. wrote a draft of the manuscript, and S.I. and Y.I. substantively revised it. All authors have read and agreed to the published version of the manuscript.

**Funding:** This work was supported by a Grant-in-Aid for Japan Society for the Promotion of Science (JSPS) Fellows (JP20J10898 to S.S.) and by JSPS KAKENHI Grant Number JP19K22289 to Y.I. and JP21K05394 to S.I.

**Acknowledgments:** We thank the Center for Advanced instrumental and Educational Supports (Faculty of Agriculture, Kyushu University) for using an image analyzer.

**Conflicts of Interest:** The authors claim that there is no conflict of interest.

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
