# Peer review of "Genetic and Biochemical Characterizations of aLhr1 Helicase in the Thermophilic Crenarchaeon Sulfolobus acidocaldarius"

_catalysts, doi:10.3390/catal12010034_

Round 1
Reviewer 1 Report
The paper presents the genetic and biochemical characterization of a novel helicase from Sulfolobus acidocaldarius. The manuscript is well-written and the study had clearly laid the foundation for further discovering whether SSB interacts and/or colocalizes with SacaLhr1.
Author Response
Dear Reviewer 1,
Thank you very much for your very positive reading of our manuscript.
We asked a native check for our English language to one of the most popular company named “editage” before submission of our original manuscript. However, according to your suggestion, we asked the company to check our English again. We hope the English language is now enough to be read.
Sincerely,
Reviewer 2 Report
In the manuscript, entitled 'Genetic and Biochemical Characterizations of aLhr1 Helicase in the Thermophilic Crenarchaeon Sulfolobus acidocaldarius' Suzuki et al. tried to understand the homologous recombination pathway in archaea by isolating a novel helicase encoded by Saci_0814, from the thermophilic crenarchaeon Sulfolobus acidocaldariusa as a model system. The paper is very good and very detailed. They were very careful in designing the experiments and writing the paper. Just few minor comments:
-Page 6, fig 2, I feel they should represent the 0:400 as –ve control.
-Page 9, Figure 4A and C and D. This figure is confusing for me. I think they should represent what is the product (p) exactly using figure like they did in fig 5?
Author Response
Dear Reviewer 2
Thank you very much for your very positive reading of our manuscript.
-Page 6, fig 2, I feel they should represent the 0:400 as –ve control.
We revised Fig 2 according to your suggestion. "0:400" was changed to "no protein". “Reactions carried out without SacaLhr1, in the presence of 900 pmol ss/dsDNA (indicated as no protein) were considered as the negative control.” was added to the legend to this figure.
-Page 9, Figure 4A and C and D. This figure is confusing for me. I think they should represent what is the product (p) exactly using figure like they did in fig 5.
We revised Figs 3 as well as Fig. 4 according to your suggestion.
Sincerely,
Reviewer 3 Report
This manuscript is mainly about identification and evidence of a gene/enzyme that is responsible for homologous recombination between DNA duplexes. Isolated from Sulfolobus acidocaldarius, Saci_0814 was first proved to be vital for homologous recombination process in vivo. Then Saci_0814 was classified as an aLhr1 under superfamily 2 helicases which shows branch migration activity in vitro. Mainly based on these two evidences, the manuscript evidences that aLhr1 is responsible for homologous recombination in vivo. In general, I think the manuscript can be published. However, I have one major concern/suggestion that should be addressed: Is it possible for authors to compare the gene sequence similarity as well as protein structure similarity of Saci_0814 with other well-studied aLhr1 helicase? This will strengthen the main conclusion of the manuscript.
Author Response
Dear Reviewer 3,
Thank you very much for your very positive reading of our manuscript.
According to your comment, we added one paragraph describing the sequence and 3D structure of Saci_0814 as compared with other Lhr helicases as follows;
“The structure of aLhr1 has been partially predicted and it is known to have a C-terminal domain containing a Zn-finger-like motif and helix-turn-helix motif (784-911 aa), which are unique to aLhr1 and not present in other Lhr family proteins. The crystal structure of the Mycobacterium smegmatis Lhr core domain (1-856 aa) in complex with AMPPNP, Mg2+, and ssDNA has been elucidated [52]. This domain possesses the helicase and translocase activities, and its structure is conserved in the aLhr2 and Hel308/Hjm from M. thermoautotrophicus [23,53]. These proteins primarily in stalled replication fork repair. Since the 3D structure of aLhr1 has not yet been determined, it is difficult to precisely predict the structure-function relationship. However, we speculate that, the unique structural features of the C-terminal domain of aLhr1 might be responsible for its affinity for specific DNA structures. Due to the differences in structure of aLhr1, its affinity towards preferred DNA substrates also varies from other Lhr family helicases.” in page 15.
Sincerely,
To add this paragraph, two references [52] and [53] were added.